# Comprehensive Clinical Genetics, Molecular and Pathological Evaluation Efficiently Assist Diagnostics and Therapy Selection in Breast Cancer Patients with Hereditary Genetic Background

**DOI:** 10.3390/ijms252312546

**Published:** 2024-11-22

**Authors:** Petra Nagy, János Papp, Vince Kornél Grolmusz, Anikó Bozsik, Tímea Pócza, Edit Oláh, Attila Patócs, Henriett Butz

**Affiliations:** 1Department of Molecular Genetics and The National Tumour Biology Laboratory, National Institute of Oncology, Comprehensive Cancer Centre, Ráth György u. 7-9, 1122 Budapest, Hungary; nagy.petra@oncol.hu (P.N.); papp.janos@oncol.hu (J.P.); grolmusz.vince@oncol.hu (V.K.G.); bozsik.aniko@oncol.hu (A.B.); pocza.timea@oncol.hu (T.P.); olah.edit@oncol.hu (E.O.); patocs.attila@oncol.hu (A.P.); 2HUN-REN-SE Hereditary Tumours Research Group, Hungarian Research Network, Nagyvárad tér 4, 1089 Budapest, Hungary; 3Department of Laboratory Medicine, Semmelweis University, Nagyvárad tér 4, 1089 Budapest, Hungary; 4Department of Oncology Biobank, National Institute of Oncology, Comprehensive Cancer Centre, Ráth György u. 7-9, 1122 Budapest, Hungary

**Keywords:** multigene panel, NGS, hereditary, germline, diagnostics, genetic testing, breast cancer, ovarian cancer

## Abstract

Using multigene panel testing for the diagnostic evaluation of patients with hereditary breast and ovarian cancer (HBOC) syndrome often identifies clinically actionable variants in genes with varying levels of penetrance. High-penetrance genes (*BRCA1*, *BRCA2*, *CDH1*, *PALB2*, *PTEN*, *STK11*, *TP53*) inform specific clinical surveillance and therapeutic decisions, while recommendations for moderate-penetrance genes (*ATM*, *BARD1*, *BRIP1*, *CHEK2*, *MLH1*, *MSH2*, *MSH6*, *PMS2*, *EPCAM*, *NF1*, *RAD51C*, *RAD51D*) are more limited. A detailed disease history, including pedigree data, helps formulate the most appropriate and personalised management strategies. In this study, we evaluated the clinical benefits of comprehensive hereditary cancer gene panel testing and a pre-sent questionnaire in Hungarian patients with suspected HBOC syndrome. We prospectively enrolled 513 patients referred for HBOC testing. Of these, 463 met the genetic testing criteria, while 50 did not but were tested due to potential therapeutic indications. Additionally, a retrospective cohort of 47 patients who met the testing criteria but had previously only been tested for *BRCA1/2* was also analysed. Among the 463 patients in the prospective cohort, 96 (20.7%) harboured pathogenic/likely pathogenic (P/LP) variants—67 in high-penetrance genes and 29 in moderate-penetrance genes. This ratio was similar in the retrospective cohort (6/47; 12.7%). In patients who did not meet the testing criteria, no mutations in high-penetrance genes were found, and only 3 of 50 (6%) harboured P/LP variants in moderate-penetrance genes. Secondary findings (P/LP variants in non-HBOC-associated genes) were identified in two patients. In the prospective cohort, P/LP variants in *BRCA1* and *BRCA2* were the most prevalent (56/96; 58.3%), and the extended testing doubled the P/LP detection ratio. Among moderate-penetrance genes, five cases (three in the prospective and two in the retrospective cohorts) had P/LP variants in Lynch syndrome-associated genes. Further immunohistochemistry analysis of breast tumour tissues helped clarify the causative role of these variants. Comprehensive clinical and molecular genetic evaluation is beneficial for the diagnosis and management of patients with P/LP variants in hereditary tumour-predisposing genes and can serve as a basis for effective therapy selection, such as PARP inhibitors or immunotherapy.

## 1. Introduction

With the advancement of molecular genetics, personalised or precision medicine has become an important goal to reach and follow. In oncology, patients with pathogenic/likely pathogenic variants in *BRCA1/2* or mismatch repair (MMR) genes would benefit from targeted therapies like poly-ADP ribose polymerase (PARP) inhibitors and immunotherapies, respectively [1]. Besides therapy, identifying a hereditary disease-causing variant can indicate participation in special surveillance programs, potential preventive options, or family screening according to international guidelines (e.g., National Comprehensive Cancer Network, NCCN) [2,3].

Currently, breast cancer (BC) is among the most frequent tumour types in women worldwide, and its incidence is increasing globally [4]. In Hungary, the standardised breast cancer incidence rate increased from 66.9 per 100,000 in 2001 to 87.1 per 100,000 in 2015, while the number of deaths associated with breast cancers was reduced (R = −0.82, *p* < 0.001) [5], perhaps due to modern diagnostics and effective therapies.

The majority of breast cancer cases are sporadic (75–80%), while approximately 15–20% occur in families with one or more affected first- or second-degree relatives, likely due to multifactorial and polygenic factors. About 5–10% of patients have monogenic germline genetic alterations [2]. Nevertheless, it is estimated that all the currently known breast cancer susceptibility genes account for less than 25% of the familial aggregation of breast cancer [6,7,8]. Besides *BRCA1* and *BRCA2*, further genes have been associated with hereditary breast and ovarian cancer (HBOC) [9,10]. In the current NCCN guidelines [2], 20 genes are listed as having clinical consequences for affecting therapy, prevention, and surveillance. Indeed, next-generation sequencing allows the testing of multiple genes at the same time at a relatively low cost with a short turnaround time, which is particularly important for cancer patients. With the application of this new technology, the investigation of all moderate-to-high penetrance genes associated with breast cancer has become routine in clinical genetic laboratories.

In our institute, “complex genetic care”, including genetic counselling and genetic testing, is provided in the same department for patients with suspected hereditary predisposition for cancer. Patients with all tumour types (including breast, ovarian, and colorectal cancer, endocrine, or other rare tumours) are tested when the suspicion of genetic predisposition (young age of onset, multiple tumour, familial appearance) is raised by the clinical oncologist and clinical geneticist according to international guidelines [2,11,12]. First, we applied targeted testing (e.g., *BRCA1/2* in breast and ovarian cancer or *MLH1*, *MSH2*, *MSH6*, and *PMS2* in Lynch syndrome suspected patients). In 2021 September, a comprehensive hereditary cancer panel was introduced to cover 113 susceptibility genes for solid tumours [13,14]. This approach offers a more efficient, faster, and more affordable solution to support oncology care compared to tumour- or syndrome-specific gene sets. Applying a wider panel could increase the diagnostic rate [15,16] and also reveal secondary findings (disease-causing genetic variants not associated with the particular disease but important for others); therefore, more patients and families can benefit from gene-specific surveillance and prevention.

Our goal was to first investigate the clinical usefulness of a comprehensive HBOC multigene panel in Hungarian patients with breast, ovarian, prostate, or pancreatic cancer, supplemented by a questionnaire on personal and family medical history.

## 2. Results

### 2.1. Detection Rate and Spectrum of Pathogenic/Likely Pathogenic Variants in HBOC-Associated Genes in Hungarian Patients

Pre-sending the questionnaire regarding the proband and family history gave time for the patients to gather all relevant medical information, which made the pre-test genetic counselling effective, and these data could be integrated during variant interpretation as well. In the prospective cohort (Table 1), out of 463 probands meeting NCCN 2021.2 criteria, a total of 96 individuals (20.7%) carried a pathogenic/likely pathogenic variant in genes associated with HBOC syndrome.

These variants span over 17 genes (*ATM*, *BARD1*, *BRCA1*, *BRCA2*, *BRIP1*, *CDH1*, *CHEK2*, *CDKN2A*, *MLH1*, *MSH2*, *NF1*, *PALB2*, *PMS2*, *PTEN*, *RAD51C*, *RAD51D*, and *TP53*) associated with elevated risk for HBOC tumours according to NCCN guidelines (Figure 1, Table 2, Appendix A).

The most commonly identified alterations were found in *BRCA1* and *BRCA2* (Figure 1). Their variants accounted for 58.3% (56/96) of all P/LP variants. Therefore, the testing of an extended breast cancer panel doubled the detection rate compared to *BRCA1/2* testing only: 20.7% (96/463) vs. 12.1% (56/463).

*CHEK2* disease-causing variants (excluding common low-penetrance variants: NM_007194.4:c.470T>C p.I157T; c.1283C>T p.S428F; and c.1427C>T p.T476M) were detected in 10 patients (10.4%; 10/96) followed by *PALB2* (7.3%; 7/96) and *ATM* (6.3%; 6/96) alterations (Figure 1). The most frequent types of variants were frameshift (43.8%), followed by missense (21.9%) and nonsense (18.8%) variants. Splice variants accounted for 10.1% of all identified mutations (Figure 1). Among all P/LP variants, CNVs were identified in eight cases: four in *BRCA1*, two in *CHEK2,* and two in the *PALB2* gene.

Four P/LP *TP53* variants were also identified (two in the prospective and two in the retrospective cohorts). In one case, clonal hemopoiesis was suspected (Figure 1 and Table 2): a female breast cancer patient with bilateral disease at ages 56 and 63 and no family history of breast cancer. A second-degree relative had gastric cancer at 68, but no medical records were available. Given the patient’s age, previous chemotherapy, and family history, clonal hemopoiesis was suspected and explained during genetic counselling. Additional examinations and family screening were offered, but the patient and her family declined further sampling and studies. The other three *TP53* cases can be considered as attenuated Li-Fraumeni syndrome cases based on the classification system of Kratz et al. (P/LP germline *TP53* variant + (history of) cancer but not meeting LFS testing criteria + no cancer before age 18 y) [17].

Regarding the seven cases with mismatch repair (MMR) gene P/LP variants, we performed immunohistochemistry staining of the breast tumour tissues. MMR deficiency was observed in four of the five samples available for testing, suggesting their causative role in the development of breast cancer (Appendix A).

Among the patients not meeting the NCCN criteria, three P/LP variants were identified (6%, 3/50) in moderate-penetrance breast cancer susceptibility genes (*CHEK2* and *ATM*) and Lynch syndrome-associated genes (*MSH2*) (see details in Appendix A).

The detection rates of VUS in HBOC genes were identical (28%) in participants regardless of meeting (130/463) or not meeting (14/50) the criteria for genetic testing (Figure 2a,b).

We could not find differences in the age of onset, tumour histology, or Ki-67 index between patients harbouring one or more VUS variants and patients with completely normal genotypes regarding HBOC genes. Similarly, the number of HBOC-associated neoplasms in their family members did not differ from wild-type carriers.

In our retrospective, *BRCA1/BRCA2* negative cohort, six cases harboured P/LP variants (6/47; 12.7%), three in high- (*PTEN*, *TP53*) and three in moderate- (*BARD1*, *MSH2*, *MSH6)* penetrance genes. (Table 2). This ratio was similar to patients meeting the testing criteria for HBOC.

### 2.2. Double Mutations in HBOC Predisposition Genes Among Breast and Ovarian Cancer Patients

Of the 96 patients, 5 carried P/LP variants in 2 HBOC-associated genes simultaneously (5.2%; 5/96) (Table 3, Figure 3, Appendix A). Among these, four patients harboured their genetic alterations in different genes, and in one case, the two P/LP variants were detected in the same gene (*CHEK2*). Family members were available for cascade testing in only two cases (Figure 3), providing insufficient data to draw solid conclusions regarding genotype–phenotype associations in individual families. Also, we could not find any difference in the age of onset, tumour histology, or Ki-67 index and the number of HBOC-associated neoplasms among family members between patients harbouring one or two P/LP variants in HBOC-associated genes.

### 2.3. Association of P/LP Variants with Clinicopathological Parameters and Family History of Cancer

High-penetrance (*BRCA1*, *BRCA2*, *CDH1*, *PALB2*, *PTEN*, *TP53*) and moderate-penetrance (*ATM*, *BARD1*, *BRIP1*, *CHEK2*, *MLH1*, *MSH2*, *MSH6*, *NF1*, *PMS2*, *RAD51C*, *RAD51D*) tumour susceptibility genes were selected according to NCCN Genetic/Familial High-Risk Assessment: Breast, Ovarian, and Pancreatic 2023.3 guideline.

We observed previously described genotype–phenotype associations such as *BRCA1* carrier status and triple-negative histology, or *BRCA1* carrier status and the frequency of multiple and/or bilateral HBOC syndrome-associated cancer (Appendix A).

The NCCN guidelines for breast cancer susceptibility gene testing outline several clinical scenarios, such as early tumour onset, multiple primary tumours, and a family history of cancer associated with HBOC syndrome. Accordingly, we cross-referenced genetic findings with criteria including young age (≤31 years), the presence of multiple primary HBOC-related cancers (either synchronous or metachronous), and family history involving more than one close relative with an HBOC-specific tumour. We investigated how many of these criteria were met in each case. We found that the detection rate in patients fulfilling one, two, or three of these criteria were 11–32, 21–75, and 100%, respectively (Figure 2b).

The median age of first tumour onset in patients with P/LP variants in genes associated with HBOC was lower than in probands with no P/LP variants (Figure 4a). Also, the mean age at the onset of the first tumour was lower in the group carrying high-penetrance mutations compared to two other groups: patients who harboured P/LP alterations in genes of moderate penetrance and probands who did not meet the criteria for genetic testing (Figure 4b).

Expectedly, we observed more HBOC-associated tumour cases among P/LP carrier proband’s family members compared to families of wild-type probands.

### 2.4. Secondary Genetic Findings

P/LP variants in genes not associated with the HBOC syndrome in patients with breast, ovarian, pancreas, or prostate cancer are considered unexpected, secondary findings. Of these, several are recommended to be reported on a diagnostic test result according to the American College of Medical Genetics and Genomics (ACMG) [18].

In our study, secondary findings were detected in two patients (0.36%; 2/560). A missense variant, (NM_020975.6):c.2410G>A, p.(Val804Met) in *RET* proto-oncogene, was identified in a female patient diagnosed with breast cancer at the age of 52. She reported one family member with breast and one with pancreatic cancer; however, no medullary thyroid cancer or other multiplex endocrine neoplasia type 2 syndrome-associated tumours could be proven in this family. In the *TMEM127* gene, a missense alteration (NM_017849.4):c.419G>A, p.(Cys140Tyr) was found in a female patient with a personal history of breast cancer diagnosed at 45 years of age presenting with one family member with breast cancer but no individual or family history for pheochromocytoma or paraganglioma.

Among ACMG reportable genes related to other tumour predispositions, the detection rate of VUS was similar to HBOC-associated genes (Figure 2a).

### 2.5. Analytical Performance of Hereditary Cancer Panel Testing

Genes associated with HBOC syndrome had an average coverage of 208 reads. Among these, the *ATM* gene had the lowest, but sufficient average coverage: 77 reads per base (Appendix A). Similar findings were seen in cancer-predisposing genes not associated with HBOC syndrome (Appendix A).

Overall, 99.6% of all bases were properly covered (>10 reads/bp) and hence clinically informative. Regarding the validated sequence regions, 100% concordance was found between the DNA sequences assessed by the two methods (technical sensitivity: 100%, specificity: 100%).

While analysing the detection of copy number variants, 215 MLPA tests were performed assessing 22 genes (*APC*, *ATM*, *BRCA1*, *BRCA2*, *CDH1*, *CHEK2*, *MLH1*, *MSH2*, *MSH6*, *MUTYH*, *NF1*, *PALB2*, *PMS2*, *PTEN*, *RAD50*, *RAD51C*, *RAD51D*, *SDHAF2*, *SDHC*, *SMAD4*, *TP53*, *TSC2*). NGS yielded a CNV call in 78 cases, of which 10 cases could be validated by MLPA. In another 137 cases, MLPA was performed to avoid false negative CNV detection; however, no CNV could be detected by MLPA. Therefore, our NGS method resulted in 100% analytical sensitivity and 67% specificity, with the negative predictive value being 100%. Our bioinformatic analysis for CNV detection is set to be overly sensitive, eliminating the possibility of false negativity, therefore confirmative testing by MLPA was non-negotiable.

## 3. Discussion

Our centre was the first in Hungary to implement a hereditary cancer gene panel testing (covering 20 genes associated with breast and ovarian cancer, according to the NCCN guideline [2]) along with genetic counselling. We evaluated its clinical utility and diagnostic rate. Pre-sending the questionnaire on the proband and family history, allowing time for patients to gather all relevant medical information, proved beneficial. This preparation facilitated effective and time-saving pre-test genetic counselling and enabled the integration of these data during variant interpretation. The analytical performance was suitable for diagnostics. We validated all identified variants by another method (Sanger sequencing or MLPA) on independent blood samples. Using an overly sensitive method ensured that we did not lose any CNV cases at the cost of numerous MLPA validations. By testing these 20 HBOC-associated genes, we were able to double the diagnostic rate (20.7% vs. 12.1%) compared to the routine procedure of testing only *BRCA1/2* genes. Our findings align with those of other studies on multigene panel testing for hereditary cancer, where the detection rate of pathogenic genetic alterations fell within a range of 10.1% to 33.8% [10,19].

The most commonly detected alterations were found in *BRCA1*, *BRCA2*, *CHEK2,* and *PALB2* genes. *BRCA1/2* are the most important breast cancer predisposition genes, representing 58.3% of all disease-causing variants, which is also in line with findings described in other populations [15,16,19]. Among disease-causing variants, the *BRCA1/2* CNV detection rate was around 10%, similar to earlier reported results [20]. After *BRCA1/2*, the *CHEK2* gene had the highest prevalence of pathogenic genetic alterations. *CHEK2* P/LP variants were found in 1.8% of all probands, accounting for 10.4% of all P/LP variants. A similar study by Bilyalov et al. on 1117 HBOC probands from Russia also reported *CHEK2* as the third most frequently affected HBOC gene (7.1%) [19]. The *CHEK2* gene has a moderate penetrance, representing a 20–40% lifetime risk for breast cancer [2]. *PALB2* is considered the third most significant breast cancer susceptibility gene after *BRCA1/2* due to its penetrance and prevalence [21]. In our study, it was the fourth most frequently altered HBOC susceptibility gene, accounting for 7.3% of all P/LP variants, which aligns with our previous findings [22]. This is similar to Hu et al.’s report of a 9.9% detection rate for *PALB2* P/LP [23]. The disease penetrance for *PALB2* P/LP patients typically falls between high and moderate [21], representing 41–60% lifetime breast cancer risk of up to 41–60% [21,24]. In contrast to *BRCA1/2*, *PALB2* P/LP variants are not currently an indication for PARP-inhibitor therapy, but their role in homologous recombination has proven beneficial for metastatic breast cancer [25]. But due to their detection ratios, P/LP variants in *BRCA1/2* and other homologue recombination repair genes represent an important indication for PARP inhibitor therapy in breast, ovarian, pancreatic, and prostate cancer [1].

*TP53* P/LP variants were identified in four patients. Clonal hemopoiesis was suspected in one case; still, the true germline P/LP *TP53* variant detection rate was higher among breast cancer patients meeting genetic testing criteria (0.43%; 2/463) compared to the control database gnomAD (0.018–0.028%) [26]. *TP53* P/LP variants have been reported to have a higher-than-expected population prevalence, corresponding to attenuated phenotypes, as classic Li-Fraumeni syndrome is rare [26,27]. *TP53* P/LP variants represent a high risk for breast cancer, even in cases with an attenuated phenotype. This can pose challenges in medical management, as there are currently no distinct surveillance protocols for patients with classic LFS, phenotypic LFS, or attenuated LFS [28]. Nevertheless, radiotherapy should be avoided due to the increased risk of post-irradiation tumours [29].

In our cohort, a germline MMR defect was identified in seven cases (three in patients meeting the HBOC genetic testing criteria, one in probands not meeting testing criteria, and two in our retrospective cohort). Five breast specimens were available for testing, and four out of the five cases showed MMR deficiency by immunohistochemistry. While some studies report an increased risk of breast cancer in *PMS2* and *MSH6* carriers [30], this could not be validated in other cohorts [31,32,33,34]. Nevertheless, a subset of breast cancer patients is etiologically linked to Lynch syndrome and may benefit from immunotherapy [35].

We detected variants of uncertain significance (VUSs) in several probands, with a prevalence of 28%, similar to other studies [36,37]. VUS diagnoses did not affect clinical management or prompt family screening, as they are often reclassified as benign over time [2,38,39,40]. Despite their current clinical irrelevance, our test report contains these results according to the recommendation by ACMG [41]. The potential outcomes and their potential reclassification are discussed during post-test genetic counselling.

When using a multigene panel, the detection of P/LP variants in more than one cancer susceptibility gene and secondary findings (genetic alteration not associated with the current disease) have relevance to clinical management. While the number of double mutations (P/LP variants in more than one HBOC gene) was not high, we could not observe a more serious clinical picture (age of onset, tumour histology, Ki-67 index, or number of HBOC-associated neoplasms among family members) in these patients compared to patients carrying only one P/LP genetic alteration. Nevertheless, both variants need to be considered in the carrier individual and during family screening. With the increased use of genetic screening (extended panels, exome, or genome sequencing), the detection of P/LP variants in two or more cancer susceptibility genes (also called MINAS, multilocus inherited neoplasia alleles syndrome) is expected to increase. Using a complex literature search and 100,000 Genomes Project data, McGuigan and Whitworth [42,43] identified 385 cases of MINAS. The authors reported that the observed clinical phenotype aligns with what we would expect if the variants of the cancer susceptibility genes were acting independently [42,43]. However, the occurrence of unusual tumour phenotypes and/or multiple primary tumours in some cases suggests the possibility of complex interactions between the cancer susceptibility genes associated with the multilocus inherited neoplasia alleles syndrome [42,43].

The identification of secondary findings in HBOC patients is considered rare [44,45]. This could be because secondary findings are most frequently detected in HBOC genes (primarily *BRCA1/2*, *PALB2*, *ATM*, and *CHEK2*), according to our findings and those of others [44]. Nevertheless, the potential and relevance of secondary findings should be discussed during pre-test counselling [46]. In our study, secondary findings were detected in high-penetrance genes associated with endocrine tumours, regarding which clinical management guidelines are available and are consistent with prior studies [44,45,46]. Hence, when the phenotype overlaps among tumour syndromes and when secondary findings are revealed, the extended panel aids the clinical management of the hereditary predisposition.

Extended genetic testing for breast cancer susceptibility genes has the potential to identify more individuals who could benefit from (i) targeted therapies, (ii) preventive measures, and (iii) cascade testing [35]. (i) Germline genetic results provide critical guidance for patients and clinicians in choosing the most appropriate surgical approach, whether breast-conserving surgery or more extensive options, such as mastectomy [35]. These results also facilitate discussions on contralateral risk-reducing mastectomy and risk-reducing salpingo-oophorectomy as preventive actions (ii) (NCCN). In systemic treatment decisions, germline pathogenic *BRCA1/2* mutations are well-established indications for PARP inhibitor therapy. Moreover, emerging evidence suggests that other genetic alterations leading to homologous recombination defects (HRD) may also become indications for PARP inhibitors in the future [47]. Additionally, breast cancers linked to hereditary MMR gene defects may be associated with MMR deficiency and could benefit from anti-PD1/PD-L1 immunotherapy [35]. (iii) For individuals with a hereditary predisposition to HBOC syndrome, participation in surveillance programs significantly enhances early detection and improves survival outcomes [2]. Identifying unaffected carriers through these programs is equally vital, as it serves as a cornerstone of cancer prevention efforts [2,35].

## 4. Methods and Materials

### 4.1. Patients and Clinical Genetic Workflow

In the workflow of the genetic services at our institute, when an oncologist suspects a hereditary condition, patients are referred for genetic counselling. During this session, the genetic counsellor reviews all available clinical data, including the patient’s personal oncology history, laboratory reports (such as pathology and histology data), and family history. If genetic testing is indicated according to national and international guidelines [2,11,12], patients are informed about the goals, uses, and benefits of the testing. Upon agreement, they sign an informed consent form. After the genetic test, results are shared with patients in a post-test counselling session, where the clinical significance of the findings is explained. In this study, we used a prospective, a retrospective, and a control cohort. A total of 513 independent patients (probands) with a personal history of hereditary breast and ovarian cancer syndrome (HBOC)-associated tumours were prospectively enrolled (Table 1 and Table 4).

These patients were assigned for genetic analysis at the Department of Molecular Genetics at the National Institute of Oncology, Budapest, Hungary between September 2021 and September 2022. Another retrospective cohort of 47 patients meeting testing criteria and previously tested for only BRCA1/2 genes was also studied (Table 1 and Table 4). There is an overlap between the analysed samples of the current study and our previous work on PALB2 genetics [22]. As an independent control cohort, population data from the Genome Aggregation Database (gnomAD v.2.1.1) were used, applying the European non-Finnish non-cancer population (n = 134,187) to compare allele frequencies, (accessed on 3 July 2023).

It is worth mentioning that we used 510 out of 1280 samples investigated in our previous work [22]. However, in this present study, we included other samples (apparently sporadic cases not meeting NCCN genetic testing criteria) as well compared to the previous work, and we performed detailed and extended genetic analysis and both proband and family history were meticulously investigated, while the previous study focused on PALB2 genetics in patients with breast and ovarian cancer patients in a prospective study [22].

Out of our prospective cohort, 463 individuals met the criteria for genetic testing according to the Hungarian Ministry of Human Capacities 2020 guideline, which follows the National Comprehensive Cancer Network (NCCN) Genetic/Familial High Risk Assessment: Breast, Ovarian, and Pancreatic guideline of 2021.2 [2]. Among the consecutively referred patients, 50 did not fulfil the testing criteria, but we included them because the NCCN guideline defines the indication for genetic testing for high-penetrance breast cancer susceptibility genes, and our aim was to assess the detection rate of moderate-penetrance breast cancer predisposition genes too and to test the incidence of P/LP variants in true sporadic cases. Clinicopathological characteristics (age of first tumour development, other manifestations, ER, PR, HER2 receptor status, and Ki67 indices) were collected from medical reports using the institutional medical informatics system and family history questionnaires.

According to Hungarian regulation, all patients received clinical genetic counselling where all aspects of the molecular genetic tests were discussed (pre-test counselling (http://www.hbcs.hu/uploads/jogszabaly/3278/fajlok/2020_EuK_20_szam_EMMI_szakmai_iranyelv_2.pdf; accessed between 21 September 2021– 4 November 2024). During genetic counselling, written informed consent was collected from all patients. As part of the Consortium of Investigators of Modifiers of BRCA1/2 (CIMBA) and the Breast Cancer Association Consortium (BCAC), standard phenotypic and epidemiological data collection were collected by a 57-question survey about personal oncology history and a 6-question survey about family history (63 questions altogether) [48,49]. Prior to genetic counselling, these questionnaires were sent to patients to gain deeper insight into the detailed phenotypes. Genetic test results were also handed over to patients in the frame of a post-test counselling session. This study was approved by the Scientific and Research Committee of the Medical Research Council of the Ministry of Health, Hungary (ETT-TUKEB 53720-4/2019/EÜIG, ETT-TUKEB 4457/2012/EKU).

### 4.2. Molecular Genetics Testing and Splice Effect Detection

Total DNA was extracted from peripheral blood samples using a Gentra Puregene Blood Kit (Cat No.: 158389, Qiagen, Hilden, Germany) according to the manufacturer’s instructions. Sequence analyses by next-generation sequencing were performed on an Illumina MiSeq or NextSeq 550Dx Instrument following library preparation using TruSight Hereditary Cancer Panel (#20029551, Illumina, San Diego, CA, USA) adhering to the manufacturer’s protocols.

For splice effect testing, RNA was isolated from blood samples collected in Tempus Blood RNA tubes (#4342792, Thermo Fisher Scientific, Waltham, MA, USA). RNA extraction was performed using Tempus™ Spin RNA Isolation Kit (#4380204, Thermo Fisher Scientific, Waltham, MA, USA). Following reverse transcription and PCR amplification, purified PCR products were analysed by Sanger sequencing (please find details in the Appendix A).

### 4.3. Molecular Genetics and Bioinformatics Analyses

Bioinformatics analysis was performed using the Illumina Dragen Enrichment pipeline (v.4.0.3, San Diego, CA, USA), assessing sequence variants and copy number alterations. All cases fulfilling our criteria were tested by CE-IVD multiplex ligation-dependent probe amplification (MLPA) assays (see below).

All pathogenic/likely pathogenic (primary and secondary) variants and variants of uncertain significance were validated by bidirectional Sanger sequencing (altogether 290 regions detecting approximately 87,000 bases) from a second DNA sample. Detection of copy number variants by MLPA was performed using SALSA MLPA probe mixes.

The clinical significance of the variants was interpreted based on The American College of Medical Genetics and Genomics (ACMG) guidelines including patient phenotype and family history [18,50,51] (please find details in the Appendix A).

### 4.4. Statistical Analyses

For the description of tumour characteristics and family history, data proportions and 95% confidence intervals were calculated using GraphPad QuickCalcs (https://www.graphpad.com/quickcalcs/ accessed between 2 January 4 November 2024).

For comparison of categorical data 2 × 2 contingency tables were used with Fisher’s Exact Test. The association of age of first tumour onset and genotype was evaluated using the Kaplan–Meier method and log-rank test. Survival curves were created with GraphPad Prism software 9.0.0. *p*-values < 0.05 were considered statistically significant.

## 5. Conclusions

Our centre first evaluated the clinical utility of an extended gene panel among patients within the HBOC tumour spectrum in Hungary. The analytical performance of our implemented test was robust, confirming its suitability for clinical diagnostics. Pre-sending the questionnaire on proband and family history, which allowed patients time to gather all relevant medical information, proved beneficial for genetic counselling and variant interpretation. The use of a hereditary cancer panel together with complementary immunohistochemistry for MMR detection doubled the diagnostic rate compared to *BRCA1/2* testing alone and identified additional mutations and secondary findings relevant to clinical management. This approach ensures more patients can potentially benefit from targeted therapy, surveillance programs, and family screening initiatives.

## Figures and Tables

**Figure 1 ijms-25-12546-f001:**
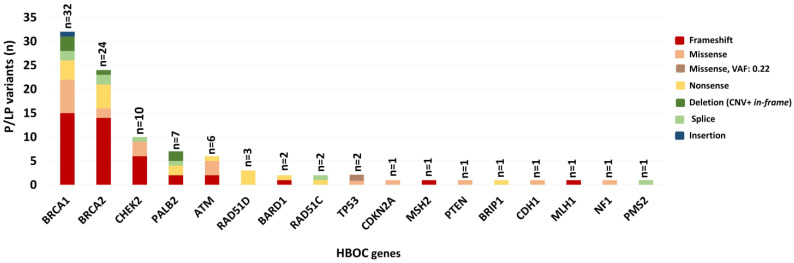
The ratio of pathogenic/likely pathogenic variants in patients meeting NCCN 2021.2 criteria and variant types.

**Figure 2 ijms-25-12546-f002:**
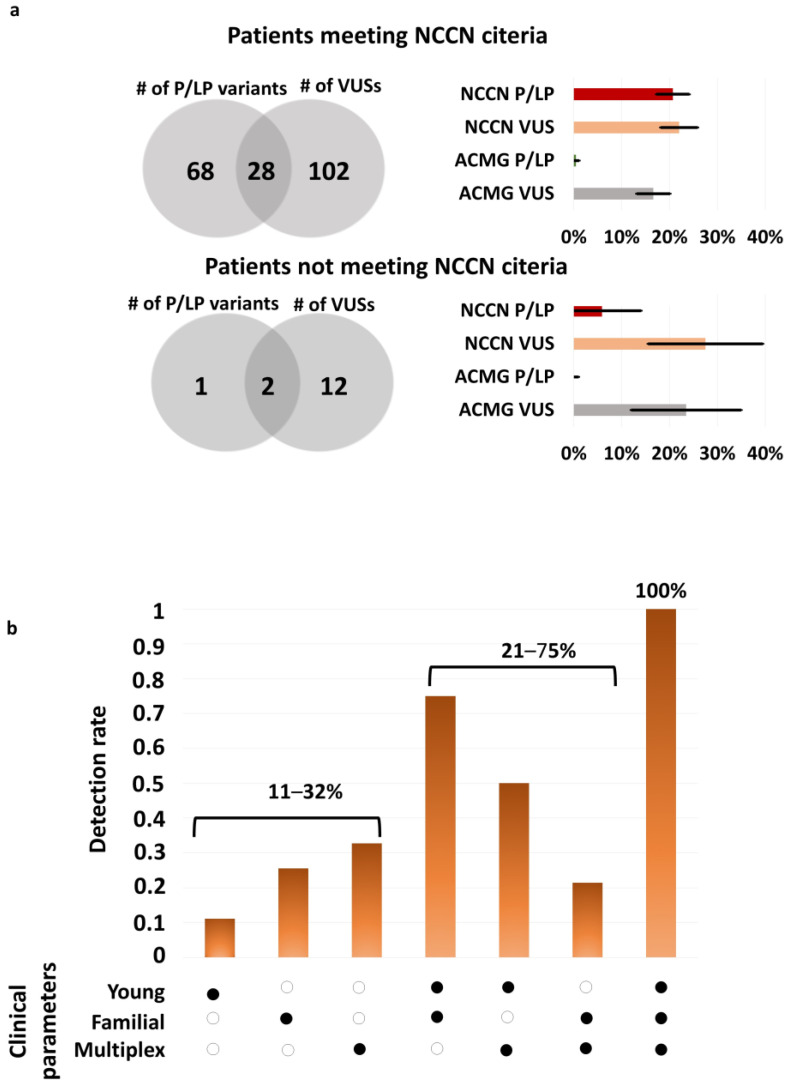
(**a**) The detection rate of variants of uncertain significance in genes associated with HBOC vs. in genes recommended to return as secondary findings according to ACMG SF v3.0 guideline [18]. (**b**) Ratio of probands harbouring P/LP variants in HBOC genes regarding clinical characteristics. #: number of variants.

**Figure 3 ijms-25-12546-f003:**
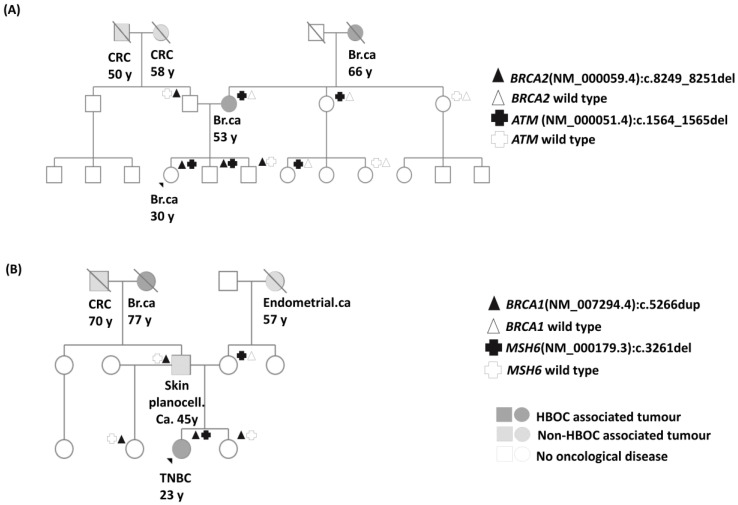
**Two** pedigrees of probands harbouring two P/LP variants (**A**,**B**) in two HBOC-associated genes where family screenings were available. (Other double-mutant cases without family screening can be found in Appendix A) Br.ca: breast cancer, CRC: colorectal cancer, HBOC: hereditary breast and ovarian cancer, TNBC: triple-negative breast cancer.

**Figure 4 ijms-25-12546-f004:**
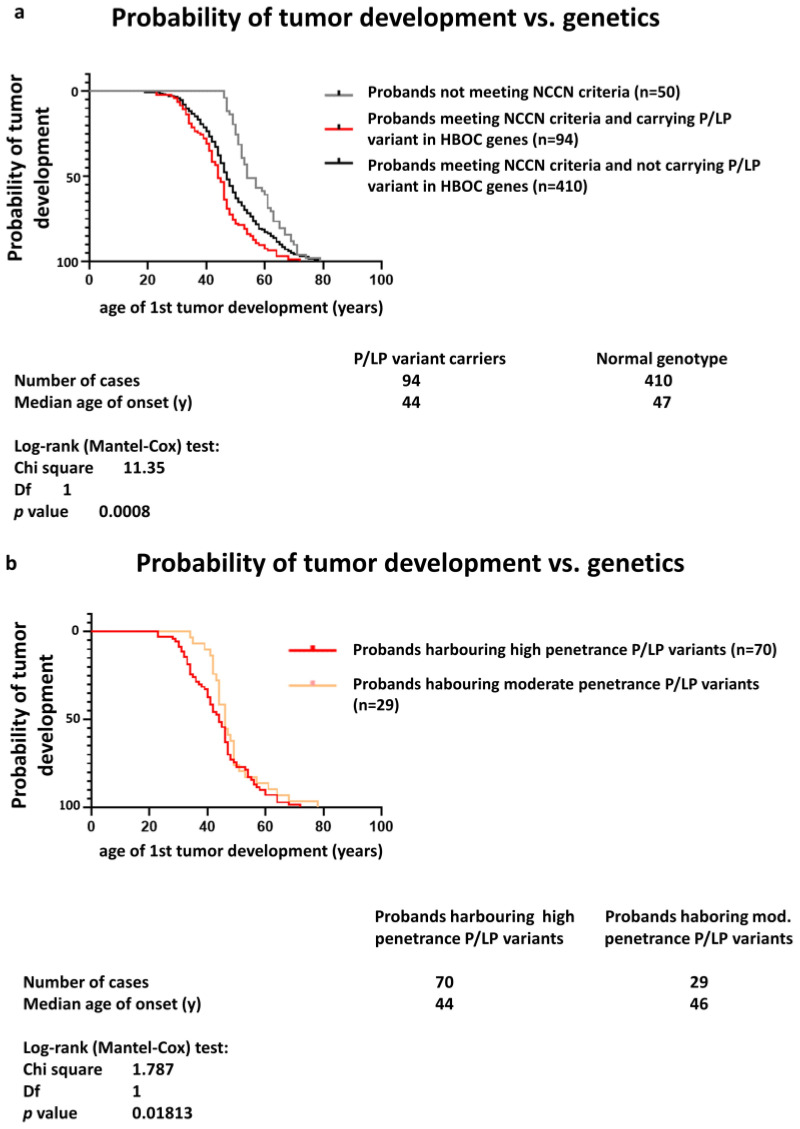
(**a**) Age at the diagnosis of the first tumour in probands with P/LP variants in HBOC genes and patients with wild-type genetic background. (**b**) Age at the diagnosis of the first tumour in probands harbouring high- vs. moderate-penetrance P/LP variants. High-penetrance genes: genes associated with >50% lifetime risk of breast cancer (*BRCA1*, *BRCA2*, *CDH1*, *PALB2*, *PTEN*, *TP53*) according to [2]; moderate-penetrance genes: genes associated with 20–50% lifetime risk of breast cancer (*ATM*, *BARD1*, *BRIP1*, *CHEK2*, *MLH1*, *MSH2*, *MSH6*, *PMS2*, *NF1*, *RAD51C*, *RAD51D*) according to [2]. Curves were calculated using all patients’ data meeting NCCN criteria, irrespective of cohorts.

**Table 1 ijms-25-12546-t001:** Demographic summary of the study population.

	Prospective Cohort				Retrospective Cohort	
	Probands Meeting NCCN 2021.2 Criteria for HBOC		Probands Not Meeting NCCN 2021.2 Criteria for HBOC			
	Number (n)	Percentage (%)	Number (n)	Percentage (%)	Number (n)	Percentage (%)
Gender						
Female	440	95.03%	46	92%	47	100%
Male	23	4.97%	4	8%	0	0%
Cancer type						
Breast	407	87.9%	46	92%	44	93.62%
Male breast	6	1.3%	0	0%	0	0%
Ovarian	26	5.62%	0	0%	3	6.38%
Pancreatic	15	3.24%	0	0%	0	0%
Prostate	9	1.94%	4	8%	0	0%
Age of onset						
≤40 years	107	23.11%	0	0%	19	40.43%
>40 years	355	76.67%	50	100%	28	59.57%
not available	1	0.22%	0	0%	0	0%

Abbreviations: NCCN: National Comprehensive Cancer Network; HBOC: hereditary breast and ovarian cancer.

**Table 2 ijms-25-12546-t002:** Identified pathogenic/likely pathogenic variants.

Gene Symbol	Variant Name HGVS, cDNA §	Variant Name HGVS, Protein	Variant Type	NCBI ClinVar Database Class	ACMG Classification	Applied ACMG Criteria	Number of Patients	Variant Allele Frequency (per Patient)
*ATM*	c.1564_1565del	p.(Glu522Ilefs*43)	FS	P	P	PM3, PVS1, PM2, PP5	1	0.54
*ATM*	c.5318del	p.(Lys1773Serfs*3)	FS	P	P	PVS1, PM2, PP5	1	0.50
*ATM*	c.5932G>T	p.(Glu1978*)	NS	P	P	PM3, PS3, PVS1, PM2, PP5	1	0.40
*ATM*	c.6095G>A	p.(Arg2032Lys)	Mis	P/LP	P	PM3, PS3, PM2, PP3, PP5	3	0.52–0.55
*ATM*	c.7096G>T	p.(Glu2366*)	NS	P/LP	P	PM3, PS3, PVS1, PM2, PP5	1	0.42
*BARD1*	c.1690C>T ∆	p.(Gln564*)	NS	Confl. (15 P; 1 VUS)	P	PS4, PS3, PVS1, PM2, PP5	2	0.41–0.38
*BARD1*	c.1932_1933del	p.(Cys645*)	FS	P/LP	P	PS4, PVS1, PM2, PP5	1	0.51
*BARD1*	c.2300_2301del	p.(Val767Aspfs*4)	FS	P/LP	P	PS3, PS4, PVS1, PM2, PP5	1	0.54
*BRCA1*	c.181T>G	p.(Cys61Gly)	Mis	P	P	PS3, PS4, PP1, PM2, PM5, PP3, PM1, PP5	7	0.40–0.56
*BRCA1*	c.3700_3704del	p.(Val1234Glnfs*8)	FS	P	P	PS4, PP1, PVS1, PM2, PP5	1	0.50
*BRCA1*	c.3756_3759del	p.(Ser1253Argfs*10)	FS	P	P	PS4, PVS1, PM2, PP5	1	0.49
*BRCA1*	c.3901_3902del	p.(Ser1301*)	NS	P	P	PS4, PVS1, PM2, PP5	1	0.50
*BRCA1*	c.3968_3971del	p.(Gln1323Argfs*12)	FS	P	P	PS4, PVS1, PM2, PP5	1	0.47
*BRCA1*	c.416dup	p.(Ser140Glufs*2)	FS	P	P	PVS1, PM2, PP5	1	0.58
*BRCA1*	c.4986+4A>T	p.(?)	Splice	P/LP	P	PS3, PS4, PM2, PP3, PP5	1	0.55
*BRCA1*	c.5251C>T	p.(Arg1751*)	NS	P	P	PS3, PS4, PVS1, PM2, PP5	2	0.40–0.50
*BRCA1*	c.5266dup	p.(Gln1756Profs*74)	FS	P	P	PS4, PS3, PVS1, PM2, PP5	10	0.44–0.56
*BRCA1*	c.5346G>A	p.(Trp1782*)	NS	P	P	PS3, PS4, PVS1, PM2, PP5	1	0.44
*BRCA1*	c.5407-1G>A	p.(?)	Splice	P	P	PS3, PS4, PVS1, PM2, PP5	1	0.47
*BRCA1*	c.68_69del	p.(Glu23Valfs*17)	FS	P	P	PS4, PS3, PVS1, PM2, PP5	1	0.61
*BRCA2*	c.1542_1543del	p.(Glu514Aspfs*12)	FS	no data	LP	PVS1, PM2	1	0.52
*BRCA2*	c.1813dup	p.(Ile605Asnfs*11)	FS	P	P	PM3, PVS1, PM2, PP5	1	0.52
*BRCA2*	c.2808_2811del	p.(Ala938Profs*21)	FS	P	P	PS4, PVS1, PS2, PM2, PP5	2	0.42–0.46
*BRCA2*	c.3483dup	p.(Ala1162Cysfs*2)	FS	no data	LP	PVS1, PM2	1	0.29
*BRCA2*	c.3975_3978dup	p.(Ala1327Cysfs*4)	FS	P	P	PS4, PVS1, PM2, PP5	1	0.48
*BRCA2*	c.5145_5146del	p.(Tyr1716*)	FS	P	LP	PVS1, PM2	1	0.53
*BRCA2*	c.5682C>G	p.(Tyr1894*)	NS	P	P	PM3, PVS1, PM2, PP5	1	0.43
*BRCA2*	c.5934dup	p.(Ser1979*)	NS	P	P	PS4, PVS1, PM2, PP5	1	0.41
*BRCA2*	c.5946del	p.(Ser1982Argfs*22)	FS	P	P	PS4, PS3, PVS1, PM2, PP5	1	0.47
*BRCA2*	c.658_659del	p.(Val220Ilefs*4)	FS	P	P	PM3, PVS1, PM2, PP5	2	0.43–0.35
*BRCA2*	c.7806-2A>G	p.(?)	Splice	P	P	PS4, PP1, PVS1, PM2, PP5	1	0.43
*BRCA2*	c.7913_7917del	p.(Phe2638*)	NS	P	P	PS4, PP1, PVS1, PM2, PP5	1	0.40
*BRCA2*	c.8168A>T	p.(Asp2723Val)	Mis	P/LP	P	PS3, PS4, PM2, PM5, PP3, PM1, PP5	1	0.41
*BRCA2*	c.8249_8251del	p.(Lys2750del)	DEL	VUS	LP	PM2, PM4, PM1	2	0.38–0.41
*BRCA2*	c.8378G>A	p.(Gly2793Glu)	Mis	Confl. (2P, 3 LP, 2 VUS)	P	PS3, PS4, PM2, PM5, PM1, PP3, PP5	1	0.54
*BRCA2*	c.9097dup	p.(Thr3033Asnfs*11)	FS	P	P	PM3, PVS1, PM2, PP5	1	0.65
*BRCA2*	c.9117G>A	p.(Pro3039=)	Splice	P	LP	PS4, PS3, PM2, PP3, PP5	1	0.48
*BRCA2*	c.9148C>T	p.(Gln3050*)	NS	P	P	PS4, PVS1, PM2, PP5	1	0.49
*BRCA2*	c.9382C>T	p.(Arg3128*)	NS	P	P	PS4, PVS1, PM2, PP5	1	0.47
*BRCA2*	c.9403del	p.(Leu3135Phefs*28)	FS	P	P	PS4, PVS1, PM2, PP5	2	0.60–0.44
*BRIP1*	c.889A>T	p.(Lys297*)	NS	no data	LP	PVS1, PM2	1	0.56
*BRIP1*	c.3525dup	p.(Ile1176Tyrfs*13)	FS	VUS	LP	PVS1, PM2, PP5	1	0.49
*CDH1*	c.1901C>T	p.(Ala634Val)	Mis	P/LP	LP	PS4, PP1, PS3, PM2, PP5	1	0.46
*CDKN2A*	c.71G>C	p.(Arg24Pro)	Mis	P	LP	PS4, PP1, PS3, PM2, PP5	1	0.45
*CHEK2*	c.1100del	p.(Thr367Metfs*15)	FS	Confl. (38 P, 1 VUS)	P	PS4, PVS1, PM2, PP5	4	0.47–0.55
*CHEK2*	c.1139_1140del	p.(Leu380fs)	FS	P	P	PS4, PVS1, PM2, PP5	1	0.68
*CHEK2*	c.499G>A	p.(Gly167Arg)	Mis	Confl. (12 LP, 1 VUS)	P	PM3, PP1, PS3, PS1, PP3, PM2, PP5	1	0.47
*CHEK2*	c.1421G>A	p.(Arg474His)	Mis	Confl. (2 LP, 6 VUS)	P	PS4, PM2, PM5, PM1, PP3, PP5	1	0.36
*CHEK2*	c.277del	p.(Trp93Glyfs*17)	FS	P	P	PS4, PS3, PVS1, PM2, PP5	1	0.44
*CHEK2*	c.434G>A	p.(Arg145Gln)	Mis	VUS	LP	PM2,PM1,PM5	1	0.44
*CHEK2*	c.444+1G>A	p.(?)	Splice	Confl. (27 P, 2 LP, 1 VUS)	P	PM3, PS3, PVS1, PM2, PP5	1	0.50
*MLH1*	c.870dup	p.(Phe291Ilefs*16)	FS	no data	LP	PVS1, PM2	1	0.43
*MSH2*	c.1226_1227del	p.(Gln409Argfs*7)	FS	P	P	PS4, PVS1, PM2, PP5	1	0.54
*MSH2*	c.586C>T ∆	p.(Pro196Ser)	Mis	Confl. (2 VUS, 1 LB)	LP	PP3, PM2, BP6	1	0.51
*MSH2*	c.873_876del	p.(Thr292Leufs*8)	FS	P	P	PS4, PVS1, PM2, PP5	1	0.40
*MSH6*	c.3261dup ∆	p.(Phe1088Leufs*5)	FS	P	P	PM3, PP1, PVS1, PM2, PP5	1	0.47
*PMS2*	c.903+3A>G ¥	p.(?)	Splice	VUS	LP ¥	PM2, PP3, PS3 ¥	1	0.44
*NF1*	c.3581A>G	p.(Asp1194Gly)	Mis	no data	LP	PM2,PP2,PM1,PP3	1	0.36
*PALB2*	c.109-2A>G	p.(?)	Splice	LP	P	PS4, PVS1, PM2, PP5	1	0.33
*PALB2*	c.1369G>T	p.(Glu457*)	NS	P	P	PVS1, PM2, PP5	1	0.31
*PALB2*	c.2336C>A	p.(Ser779*)	NS	P/LP	LP	PVS1, PM2	1	0.43
*PALB2*	c.509_510del	p.(Arg170Ilefs*14)	FS	P	P	PS4, PS3, PVS1, PM2, PP5	2	0.58–0.51
*PTEN*	c.413A>G	p.(Tyr138Cys)	Mis	VUS	LP	PM1, PP2, PM2, PP3	1	0.46
*PTEN*	c.493-1G>A ∆	p.(?)	Splice	P	P	PS4, PVS1, PM2, PP5	1	0.59
*RAD51C*	c.405-1G>A	p.(?)	Splice	LP	P	PVS1, PM2, PP5	1	0.47
*RAD51C*	c.955C>T	p.(Arg319*)	NS	P	P	PS4, PVS1, PM2, PP5	1	0.67
*RAD51D*	c.556C>T	p.(Arg186*)	NS	P	P	PS4, PVS1, PM2, PP5	1	0.26
*RAD51D*	c.757C>T	p.(Arg253*)	NS	P	P	PS4, PVS1, PM2, PP5	1	0.50
*RAD51D*	c.898del	p.(Arg300Aspfs*10)	NS	LP	LP	PVS1, PM2, PP5	1	0.46
*RET* †	c.2410G>A	p.(Val804Met)	Mis	P/LP	P	PS4, PP1, PS3, PM2, PM5, PP3, PP5	1	0.40
*TMEM127* † $	c.419G>A	p.(Cys140Tyr)	Mis	VUS	P $	PM2, PP3, PM5	1	0.48
*TP53*	c.323_329dup ∆	p.(Leu111Phefs*40)	FS	P	P	PS4, PVS1, PM2, PP5	1	0.42
*TP53*	c.473G>A	p.(Arg158His)	Mis	P/LP	P	PS3, PS4, PP1, PM1, PP2, PM2, PM5, PP3, PP5	1	0.40
*TP53*	c.614A>G	p.(Tyr205Cys)	Mis	P	P	PS3, PS4, PP1, PM1, PP2, PM2, PM5, PP3, PP5	1	0.22 £
*TP53*	c.902del ∆	p.(Pro301Glnfs*44)	FS	P	P	PS4, PVS1, PM2, PP5	1	0.48
*BRCA1*	del(ex1–20)	p.(?)	DEL	P	P	1A, 2C-1, 3A, 4C, 5G	1	HZ validated by MLPA
*BRCA1*	del(ex21–22)	p.(?)	DEL	P	P	1A, 2E, 3A, 4C, 5G	1	HZ validated by MLPA
*BRCA1*	del(ex5–10)	p.(?)	DEL	P	P	1A, 2E, 3A, 4C, 5G	1	HZ validated by MLPA
*BRCA1*	dup(ex13)	p.(?)	DUP	P	P	1A, 2I, 3A, 4C, 5G	1	HZ validated by MLPA
*CHEK2*	del(ex9–10)	p.(?)	DEL	P	P	1A, 2E, 3A, 4C, 5G	2	HZ validated by MLPA
*PALB2*	del(ex9–10)	p.(?)	DEL	P	P	1A, 2E, 3A, 4C, 5G	1	HZ validated by MLPA
*PALB2*	del(ex11)	p.(?)	DEL	P	P	1A, 2E, 3A, 4C, 5G	1	HZ validated by MLPA

All sequence and copy number variations were validated by Sanger sequencing and multiplex ligation-dependent probe amplification on independent DNA samples (see details in the methods). Abbreviations: Confl: conflicting, FS: frameshift, NS: nonsense, Mis: missense, Splice: splice variant, DEL: deletion, DUP: duplication, P: pathogenic, LP: likely pathogenic, VUS: variant of uncertain significance, HZ: heterozygote, MLPA: multiplex ligation-dependent probe amplification, HGVS: nomenclature according to Human Genome Variation Society, ACMG: American College of Medical Genetics and Genomics, NCBI: National Center for Biotechnology Information. †: incidental (secondary) finding; $: Experts (European-American-Asian Pheochromocytoma-Paraganglioma Registry Study Group) classified this variant as “pathogenic”; ¥: We performed an in vitro functional assay to test the splice effect of this variant. As aberrant splicing and exon skipping were detected (PS3), this variant was classified as likely pathogenic; £: Based on the variant allele frequency, in this case, clonal hemopoiesis was suspected considering the patient’s age, previous oncological treatment and negative family history. Please find details in the text. §: HGVS description was given according to MANE transcripts; ∆ Variants identified in patients of our retrospective cohort.

**Table 3 ijms-25-12546-t003:** Genotype and phenotype of patients with two pathogenic/likely pathogenic variants in HBOC-associated genes.

Proband’s Genotype	Proband’s Phenotype	Cancer Cases in Family
*BRCA2*(NM_000059.4):c.5682C>G p.(Tyr1894Ter) *BRIP1*(NM_032043.3):c.3525dup p.(Ile176TyrfsTer13)	breast cancer at age 47	1 breast cancer at the age of 41 1 breast cancer at the age of 51
*BRCA2*(NM_000059.4):c.8249_8251del p.(Lys2750del) *ATM*(NM_000051.4):c.1564_1565del p.(Glu522IlefsTer43)	breast cancer at age 30	1 breast cancer at the age of 51 1 breast cancer > 60 years of age
*BARD1*(NM_000465.2):c.1932_1933del p.(Cys645*) *ATM*(NM_000051.4):c.6679C>T p.(Arg2227Cys)	breast cancer at age 44	3 breast cancer cases between the ages of 51–60
*BRCA1*(NM_007294.4):c.5266dup p.(Gln1756Profs*74) *MSH6*(NM_000179.3):c.3261del p.(Phe1088Serfs*2)	breast cancer at age 23	1 breast cancer > 60 years of age
*CHEK2*(NM_007194.4):c.499G>A p.(Gly167Arg) *CHEK2*(NM_007194.4):del ex9–10 p.(?)	breast cancer at age 37	2 breast cancer cases between the ages of 51–60 years 1 breast cancer > 60 years of age

**Table 4 ijms-25-12546-t004:** Clinicopathological features of first cancers in different cohorts.

	Prospective Cohort Probands Meeting NCCN 2021.2 Criteria for HBOC	Prospective Cohort Probands Not Meeting NCCN 2021.2 Criteria for HBOC	Retrospective Cohort Probands
**Female breast cancer (n)**	407	46	44
Age of onset (years mean ± SD)	47.1 ± 11.0	56.6 ± 8.2	43.3 ± 9.8
Histology-ductal	83% (338/407)	84.8% (39/46)	79.6% (35/44)
Histology-lobular	8% (33/407)	8.7% (4/46)	6.8% (3/44)
Histology-other/mixed/unknown	9% (36/407)	6.5% (3/46)	13.6% (6/44)
LumA	25.6% (104/407)	34.8% (16/46)	43.2% (19/44)
LumB-Her2-	21.4% (87/407)	26.1% (12/46)	11.4% (5/44)
LumB-Her2+	13.0% (53/407)	15.2% (7/46)	4.5% (2/44)
Her2+	7.1% (29/407)	15.2% (7/46)	11.4% (5/44)
TNBC	25.3% (103/407)	0	9.1% (4/44)
Ki-67 index (mean ± SD)	32.7 ± 26.1	26.1 ± 20.8	22.3 ± 23.0
**Ovarian cancer (n)**	26	0	3
Age of onset (years mean ± SD)	53.4 ± 12.5	-	36.3 ± 12.3
Histology-high grade serous	65.4% (17/26)	-	33.3% (1/3)
Histology-non-serous other type	15.4% (4/26)	-	33.3% (1/3)
Histology not available	19.2% (5/26)	-	33.3% (1/3)
**Pancreatic cancer (n)**	15	0	0
Age of onset (years mean ± SD)	61.0 ± 10.0	-	-
Histology: PDAC	100% (15/15)	-	-
**Prostate cancer (n)**	9	4	0
Age of onset (years mean ± SD)	63.6 ± 4.4	65.0 ± 6.4	0
Histology-Gleason score (avg ± SD)	7.8 ± 1.1	6.0 ± 0	-
Metastatic	66.6% (6/9)	0% (0/4)	-
**Male breast cancer (n)**	6	0	0
Age of onset (years mean ± SD)	66.2 ± 9.7	-	-
Histology-ductal	100% (6/6)	-	-
Histology-lobular	0	-	-
Histology-other/mixed/unknown	0	-	-
LumA	16.7% (1/6)	-	-
LumB-Her2-	33.3% (2/6)	-	-
LumB-Her2+	33.3% (2/6)	-	-
Her2+	0	-	-
TNBC	0	-	-
Ki-67 index (mean ± SD)	35.6 ± 17.6	-	-

*Abbreviations:* LumA: luminal A subtype; LumB: luminal B subtype; TNBC: triple-negative breast cancer; Her2: receptor tyrosine-protein kinase erbB-2; SD: standard deviation; PDAC: pancreatic ductal adenocarcinoma.

## Data Availability

All data in the study are included in the article, further inquiries can be directed to the corresponding author.

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
