# Peer review of "Comprehensive Clinical Genetics, Molecular and Pathological Evaluation Efficiently Assist Diagnostics and Therapy Selection in Breast Cancer Patients with Hereditary Genetic Background"

_ijms, 2024, doi:10.3390/ijms252312546_

Round 1

Reviewer 1 Report

Comments and Suggestions for Authors

Comments and suggestions for authors

The article gives an interesting information about result of germline multi-gene panel testing of breast cancer patients in Hungarian population and it gives an insight into the spectrum of mutations in HBOC genes in this population.

Here are some suggestions for authors.

Introduction section:

Under lines 54-55: “In Hungary, the standardized incidence increased between 2001 and 2015,..«, adding numbers would be recommended.

Result section

In the sub-title “2.1. High diagnostic rate and mutational profile of pathogenic/likely pathogenic variants in HBOC-associated genes in Hungarian patients.”

“High diagnostic rate” already suggests a conclusion based on results, the authors might consider simplifying the tile, like “Detection rate and spectrum of pathogenic…

Since the table 2 is very large the authors are suggested to put it in the supplementary.

I didn’t find the Supplementary figures.

On the Figure 3 the authors should explain the (-wild type), does it mean that the tested individual is wild type for both (BRCA2 and ATM) mutations, or just for ATM, The combination of signs (*-) is confusing in the Figure 3a as well as 3b.

Under the subtitle “2.3. Association of P/LP variants with clinicopathological parameters and family history of cancer«

Lines 208 -213.- The association between the sentence “In the NCCN guidelines for testing criteria for breast cancer susceptibility genes, several clinical scenarios are defined. « and criteria named in the following text is not clear. Whose are criteria?  It is suggested that the authors would restate the paragraph.

Under Figure 4. In the underlying text, line 228; the abbreviation NNCN should probably be corrected to NCCN.

Under the subtitle “2.5. Analytical performance of hereditary cancer panel”

Lines 252-256: Since the article is focused on HBOC genes, The information about genes not associated with HBOC syndrome doesn’t seem necessary.

Authors might consider shortening the text under Methods section, … where numbering MLPA kits and bioinformatic tools and platforms used for variant interpretation. Certain parts might be moved to supplementary files.

Author Response

Reviewer 1:

We sincerely appreciate the Reviewer’s work and the overall positive opinion on our manuscript. Please find our detailed, point-by-point responses in the attached document.

Also, as per the Reviewer's comment, we have included the Supplementary Material at the end of our responses to facilitate the revision process, in addition to uploading it to the manuscript submission system.

We hope that our responses and the revised manuscript meet the Reviewer's expectations.

Sincerely,

Henriett Butz

Reviewer 2 Report

Comments and Suggestions for Authors

The authors evaluate, in this manuscript, a genetical panel with high and moderate penetrance genes for the diagnostic evaluation of patients with hereditary breast and ovarian cancer syndrome. I think that this work could be of interest for hereditary cancer patients and the general knowledge of cancer genetics. However, there is a lot of emphasis throughout the paper about the importance of this work for the institute where the authors work, and I think there should be more emphasis about cancer patients in general. I number below point by point some recommendations.

1. Table 2 is already quite long and by adding all the abbreviations and explanations below the table it ends up being very difficult to read. The information the authors present here is very relevant so there is no need to cut something out but rather could the authors consider rephrasing, restyling or shrinking the text or line spacing under the table as to make it a bit more friendly. Some explanations for the pathogenicity of the variant could perhaps be moved to the results. Also, there is the same sentence under and above the table so one could be deleted.

2. In the materials and methods section the authors should describe in more detail their patient groups. Table 1 describes gender, cancer types and divides patients into two age groups. This is acceptable for the results section as it shows their results but for the methods section it would be more suitable to show profile of their patient groups. For example, age+SD, histopathological diagnosis, in case of breast cancer were the patients mostly luminal A?  triple negative? And perhaps some more details so the patient group is well defined.

3. In the discussion (and other sections) the authors highlight the importance of this work for their center and for the diagnostic algorithm of their patients. However, they should also put some emphasis into the importance of this work for the patients in general and generally for the genetical information about the cancer patients and not only the immediate importance for their health center. The authors should rewrite or add some more information in this sense.

Author Response

Reviewer 2:

We sincerely appreciate the Reviewer’s work and the positive words on our manuscript. Please find our detailed, point-by-point responses in the attached document.

We hope that our responses and the revised manuscript will meet the Reviewer's expectations.

Sincerely,

Henriett Butz
